# TRAINING-FREE GENERALIZATION ON HETEROGENEOUS TABULAR DATA VIA META-REPRESENTATION

## ABSTRACT

Tabular data is prevalent across various machine learning domains. Yet, the inherent heterogeneities in attribute and class spaces across different tabular datasets hinder the effective sharing of knowledge, limiting a tabular model to benefit from other datasets. In this paper, we propose Tabular data Pre-Training via Meta-representation (TabPTM), which allows one tabular model pre-training on a set of heterogeneous datasets. Then, this pre-trained model can be directly applied to unseen datasets that have diverse attributes and classes *without additional training*. Specifically, TabPTM represents an instance through its distance to a fixed number of prototypes, thereby standardizing heterogeneous tabular datasets. A deep neural network is then trained to associate these meta-representations with dataset-specific classification confidences, endowing TabPTM with the ability of *training-free generalization*. Experiments validate that TabPTM achieves promising performance in new datasets, even under few-shot scenarios.

## 1 INTRODUCTION

Tabular data, with rows representing instances and columns corresponding to attributes (features), is ubiquitous in machine learning applications (Borisov et al., 2022) such as financial prediction (Cao & Tay, 2001), recommendation system (Richardson et al., 2007), and healthcare analytics (Ogunleye & Wang, 2020). Due to the inherent variability of tabular tasks, the tabular model for a given task often requires careful design or training (Hutter et al., 2019; He et al., 2021), and should be re-trained when the spaces of attributes and classes change (Hou & Zhou, 2018). While pre-trained models have achieved tremendous success in areas like natural language processing (Devlin et al., 2019; Liu et al., 2019; Zhou et al., 2023) and computer vision (Dosovitskiy et al., 2021; Kirillov et al., 2023), enabling "zero-shot" and "few-shot" generalization to new tasks (Radford et al., 2021; Alayrac et al., 2022), their experience is difficult to be extended to the tabular domain. A significant obstacle is the heterogeneity among tabular datasets — the attribute as well as class spaces often differ substantially from one dataset to another, hampering the joint training of a tabular model on multiple datasets, let alone the direct application of pre-trained models.

Some recent methods take advantage of the semantic meaning of columns. By transforming an example into textual form, the strong ability of large language models could be applied (Hegselmann et al., 2023; Wang & Sun, 2022; Liu et al., 2022a; Wang et al., 2023). Yet, these methods face limitations when column semantics are ambiguous or unavailable in real-world applications. Without the semantic meaning of attributes, some approaches either focus on adapting a tabular model from one dataset to another (Ye et al., 2021; Onishi et al., 2023), or explore shareable transformations which facilitate the tuning of remaining parameters given a downstream dataset (Iwata & Kumagai, 2020; Kumagai et al., 2022; Liu et al., 2022b; Wydmanski et al., 2023; Zhu et al., 2023). To enhance the utility of a pre-trained tabular model and save tuning resources during its deployment, we ask:

> Is it possible to pre-train a powerful model over heterogeneous tabular datasets, enabling seamless generalization to downstream tasks without additional tuning?

In this paper, we propose to represent an instance in a tabular dataset through its distance to a fixed number of prototypes from a certain class. Such a "meta-representation" effectively standardizes heterogeneous datasets, rendering them into a *uniform* form of the same dimension. By training a joint deep neural network on the meta-representations of a large number of datasets, the classi-

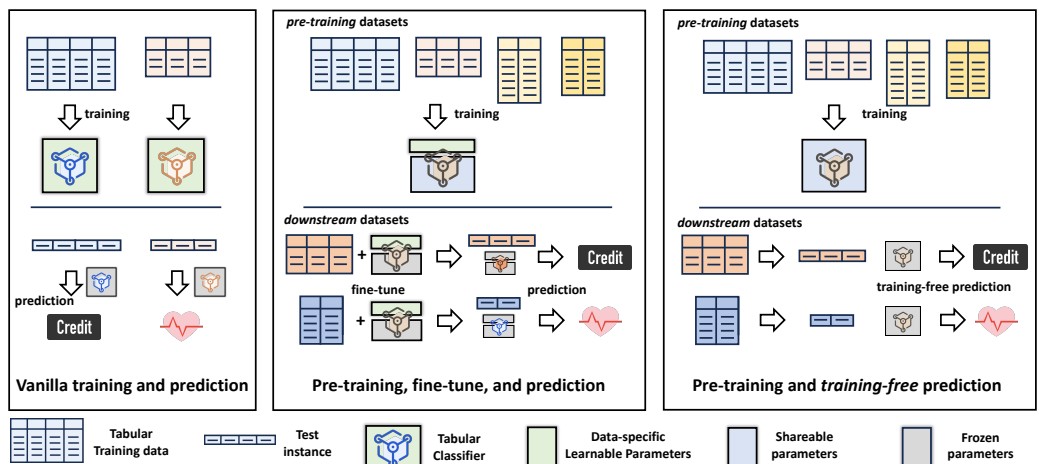

Figure 1: An illustration of training strategies on tabular data. The left one depicts the vanilla training and prediction, where tabular models are trained on each dataset separately. The middle one shows pre-training a joint model across datasets with shared parameters, and the non-shared parameters will be fine-tuned given a downstream dataset before prediction. In our training-free approach (right), a model is trained on heterogeneous tabular datasets. Then we can directly apply the pre-trained model to make predictions without additional training for a downstream dataset.

fication ability among datasets is extracted and shared. The Tabular model Pre-Trained via Meta-representation (TabPTM) could be easily extended to downstream tabular datasets in a *training-free* manner. Figure 1 compares different strategies that train and deploy models over tabular datasets.

In particular, the meta-representation characterizes an instance based on its similarity to prototypes from each class's training set given any tabular dataset. Paired with a distance metric, the meta-representation adaptively filters out redundant and noisy attributes, accurately reflecting the class membership of an instance. A deep neural network learns to map the class-wise meta-representations in each dataset to the confidence scores for classification, benefiting final predictions. Our experimental results demonstrate that TabPTM achieves promising generalization ability on unseen datasets efficiently in both few-shot and full-shot scenarios. The contributions are:

- We utilize meta-representations to reduce attribute heterogeneity and enable the pre-training of a joint classification model over tabular datasets.
- We explore how to make predictions based on the meta-representations, and the pre-trained TabPTM is capable of generalizing to unseen tabular datasets without additional training.
- Meta-representation is validated as an effective way for tabular classification. TabPTM shows promising capabilities in generalizing to unseen datasets.

## 2 RELATED WORK

**Learning with tabular data**. Tabular data is one of the most common data forms in many fields (Richardson et al., 2007; Vanschoren et al., 2014; Hamidieh, 2018), and a lot of classical machine learning methods have been developed for tabular data, such as XGBoost (Chen & Guestrin, 2016), LightGBM (Ke et al., 2017), and CatBoost (Prokhorenkova et al., 2018). Recently, researchers have tried to extend the success of deep neural networks such as multi-layer perceptron (Gorishniy et al., 2021), Transformer (Huang et al., 2020), and diffusion models (Kotelnikov et al., 2023) from visual and textual domains to the tabular fields (Borisov et al., 2022). Attribute embeddings (Song et al., 2019) and deep architectures have been designed (Guo et al., 2017; Katzir et al., 2021; Chen et al., 2023b) for tabular data, and some simple baselines can achieve competitive results as the classical methods after carefully tuned (Kadra et al., 2021). Deep tabular models have the flexibility for various scenarios and can be incorporated well with the classical methods (Cheng et al., 2016; Wang et al., 2017; Shwartz-Ziv & Armon, 2022; Grinsztajn et al., 2022).

**Reuse a heterogeneous tabular model**. Instead of training a tabular model from scratch on a new task, reusing a pre-trained model from a related task becomes a useful choice, especially when effi-

ciency is emphasized (Tommasi et al., 2014; Kuzborskij & Orabona, 2017; Aghbalou & Staerman, 2023). In addition to the distribution shift between the pre-trained and the target tasks, the changes in their attribute spaces as well as the class spaces make the transfer of a tabular model challenging (Hou & Zhou, 2018; Ye et al., 2021). The reuse of tabular models across heterogeneous datasets usually relies on some assumptions. For example, the existence of a set of overlapped attributes between two datasets (Hou et al., 2022; Levin et al., 2023; Onishi et al., 2023) and the column meanings (the textual names of all attributes) (Wang & Sun, 2022).

**Learning with multiple tabular datasets**. One more step to take advantage of the ability of deep neural networks on tabular fields is to pre-train a discriminative model on a large number of tabular datasets and extend its ability to downstream tasks. The in-distribution generalization ability of a pre-trained tabular model has been validated in multi-task learning (Argyriou et al., 2006; Zhang & Yang, 2022; Rubachev et al., 2022; Luetto et al., 2023) and self-supervised learning (Ucar et al., 2021b; Bahri et al., 2022), where all datasets are collected in a homogeneous form. In multi-view learning, a model is required to capture the consistent nature among heterogeneous views, but those multiple views of an instance are paired and share the same class space (Xu et al., 2013; Ye et al., 2015). Some recent approaches utilize the deep neural network to pre-train a more generalizable tabular model, taking the difference in attributes and classes into account. One representative kind of approach assumes the existence of attribute names along with a dataset so that each instance could be transformed into a text, then a large language model could be applied to generalize the classification ability (Liu et al., 2022a; Hegselmann et al., 2023; Zhang et al., 2023a; Wang et al., 2023). Another thread of method learns shared components such as attribute-agnostic transformation across datasets, which provides a good model initialization given a downstream task (Iwata & Kumagai, 2020; Liu et al., 2022b; Zhang et al., 2023b; Shen et al., 2023; Zhu et al., 2023). We propose TabPTM to transform all datasets into a uniform form with meta-representation to enable the pre-training. Then, the pre-trained model could be applied directly to a downstream dataset in a training-free manner.

## 3  PRELIMINARY

**Learning with a single tabular dataset**. We denote a tabular classification dataset as $\mathcal{D} = \{(\boldsymbol{x}_i, y_i)\}_{i=1}^{N}$ which has $N$ examples (rows in the table) and $C$ classes. Each instance $\boldsymbol{x}_i \in \mathcal{X}$ is depicted by $d$ attributes (columns).[1] Denote the class set of the task as $\mathcal{Y}$, the goal is to learn a tabular classifier $f$, $e.g.$, linear classifiers, decision trees, or deep neural networks, mapping an instance $\boldsymbol{x}_i$ to its label $y_i$. $f$ could be learned on the training set by minimizing the empirical objective $\min_f \sum_{i=1}^{N} \ell(f(\boldsymbol{x}_i), y_i)$, where $\ell(\cdot, \cdot)$ is a loss function that measures the discrepancy between a prediction and the label. The generalization ability of $f$ is measured by the prediction accuracy on an unseen instance sampled from the same distribution as $\mathcal{D}$.

**Learning with multiple tabular datasets**. Assume there are $T$ datasets $\mathbb{D} = \{\mathcal{D}_1, \ldots, \mathcal{D}_T\}$. The attribute and label spaces of the $t$-th dataset are $\mathcal{X}_t$ and $\mathcal{Y}_t$. The number of instances, attributes (the dimension of instance), and classes in $\mathcal{D}_t$ are denoted as $N_t$, $d_t$, and $C_t$, respectively. Different from vanilla multi-task learning where all datasets are homogeneous (Argyriou et al., 2006), $i.e.$, with the same $\mathcal{X}_t$ and $\mathcal{Y}_t$, here we consider *heterogeneous* datasets where the meaning of attributes and classes change from one dataset to another. In this case, the classifier $f$ is trained over $T$ datasets

$$\min_f \sum_{t=1}^{T} \sum_{i=1}^{N_t} \ell\left(f(\boldsymbol{x}_i^t), y_i^t\right) . \tag{1}$$

By learning on heterogeneous tasks, the model $f$ should deal with different attribute and label sets. The pre-trained $f$ then generalizes its discerning ability to an unseen task $\mathcal{D}_u$, and even in few-shot scenarios where the size $N_t$ of $\mathcal{D}_u$ is small. Assume a test instance of $\mathcal{D}_u$ is $\boldsymbol{x}_*^u$, its label could be predicted through

$$\hat{y}_*^u = f(\boldsymbol{x}_*^u \mid \mathcal{D}_u) . \tag{2}$$

In some cases, several steps of gradient descent are required to adapt the joint model $f$ on the target dataset $\mathcal{D}_u$ (Zhu et al., 2023). The *training-free* generalization means the model $f$ is able to predict the unseen instance $\boldsymbol{x}_*^u$ in task $\mathcal{D}_u$ without additional training.

---

[1]We assume all attributes of an instance are numerical (continuous). If there exist categorical (discrete) attributes, we transform them into the one-hot forms in advance.

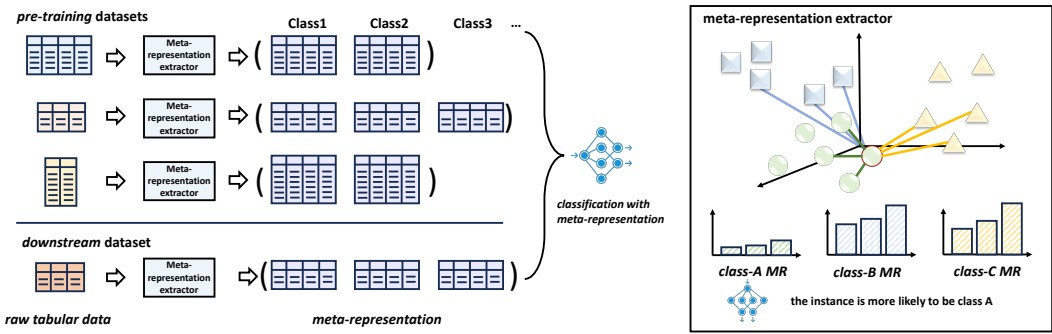

Figure 2: An illustration of the Meta-Representation (MR). MR transforms heterogeneous tabular datasets with different dimensions into a homogeneous form. A dataset has a set of $K$-dimension of MRs, one for each class. The prediction scores for different classes could be obtained via MRs. We pre-train a joint model on MRs of different datasets and extend its generalization ability to downstream datasets. The right figure shows the MR of an instance, containing distances from an instance to the nearest prototypes of a certain class, characterizes the class membership patterns.

## 4 METHOD

Considering the inherent heterogeneity in attribute and class spaces across various tabular datasets, the core idea of TabPTM is to standardize diverse datasets so that a joint deep neural network can be applied. We define an instance's membership to a class in terms of its distance to the top-$K$ nearest prototypes within the training set. This process transforms any instance, irrespective of its original dimensionality, into a set of $K$-dimensional vectors, one for each of the $C$ classes. Based on this meta-representation, a multi-layer perceptron is trained across multiple tabular datasets, which helps recognize class membership patterns and subsequently extract class-specific confidence scores. Armed with this pre-trained model, we can directly compute the meta-representation of any downstream dataset and proceed with generalization-free classification. TabPTM is illustrated in Figure 2.

### 4.1 META-REPRESENTATION OF AN INSTANCE

**Vanilla meta-representation.** First, we describe how to obtain the vanilla meta-representation for any instance in a tabular classification dataset $\mathcal{D}$ with $C$ classes. Based on the label of each instance, we collect the same-class instances in $\mathcal{D}$ together into $C$ sets:

$$\mathcal{D}_{y=c} = \{(\boldsymbol{x}_i, y_i) \mid y_i = c\}, \forall c = 1, \ldots, C . \tag{3}$$

Then we extract class-specific prototypes based on $\mathcal{D}_{y=c}$. Instances themselves are kept in Equation 3 as prototypes, and more ways to obtain prototypes will be introduced later. Given a class $c$, we calculate the distance between an instance $\boldsymbol{x}_i$ to those prototypes in $\mathcal{D}_{y=c}$ ($|\mathcal{D}_{y=c}|$ instances in Equation 3), and sort them in an *ascending* order:

$$\{\text{dist}(\boldsymbol{x}_i, \boldsymbol{x}_1), \ldots, \text{dist}(\boldsymbol{x}_i, \boldsymbol{x}_j), \ldots, \text{dist}(\boldsymbol{x}_i, \boldsymbol{x}_{|\mathcal{D}_{y=c}|})\}$$
$$\text{s.t. } \text{dist}(\boldsymbol{x}_i, \boldsymbol{x}_1) \leq \ldots \leq \text{dist}(\boldsymbol{x}_i, \boldsymbol{x}_j) \leq \ldots \leq \text{dist}(\boldsymbol{x}_i, \boldsymbol{x}_{|\mathcal{D}_{y=c}|}) . \tag{4}$$

Here $\text{dist}(\cdot, \cdot)$ measures the instance-wise distance, *e.g.*, Euclidean distance or Manhattan distance. We then select the $K$ smallest distance values in the set, which constructs the meta-representation for the instance $\boldsymbol{x}_i$. We define the mapping from $\boldsymbol{x}_i$ to its meta-representation by $\phi_c$:

$$\phi_c(\boldsymbol{x}_i) = [\text{dist}(\boldsymbol{x}_i, \boldsymbol{x}_1), \ldots, \text{dist}(\boldsymbol{x}_i, \boldsymbol{x}_j), \ldots, \text{dist}(\boldsymbol{x}_i, \boldsymbol{x}_K)] \in \mathbb{R}^K . \tag{5}$$

$\phi_c(\boldsymbol{x}_i)$ captures the neighborhood distribution of an instance, revealing its membership to a particular class. If an instance resides within a high-density region of a class (akin to being near the class center), the majority of values in $\phi_c(\boldsymbol{x}_i)$ would typically be small, indicating close proximity to neighboring instances of that class. Conversely, if only a few values in $\phi_c(\boldsymbol{x}_i)$ are small, while most are large, it indicates that the instance $\boldsymbol{x}_i$ is likely located at the boundary among classes. In summary, $\phi_c(\boldsymbol{x}_i)$ transforms any instance $\boldsymbol{x}_i$ based on its relationship with class-wise prototypes in $\mathcal{D}$. No matter what value the original dimension $d$ of $\boldsymbol{x}_i$ is, $\phi_c(\boldsymbol{x}_i)$ has a fixed dimension with value $K$, standardizing the vectors and facilitating pre-training over heterogeneous tabular datasets.

**Prototypes for each class.** Prototypes for each class are extracted based on $\mathcal{D}_{y=c}$ before calculating distances in Equation 4. Besides using all instances from the $c$-th class, we investigate several other kinds of prototypes to reveal the characteristics of a class. For example, we randomly sample a subset of the instances in $\mathcal{D}_{y=c}$, or apply some clustering approaches such as $k$-means to sketch the set. We empirically observe that although clustering helps in some cases, it may lose some details for partial tabular datasets. Keeping the full training set as in Equation 3 is beneficial in most cases, so we use it as the default option.

**Metric-based meta-representation.** The distance measure dist in Equation 4 utilizes either Euclidean or Manhattan distance. Yet, in the presence of high-dimensional features (large $d$), relying on all attributes becomes computationally challenging. Moreover, the distance might be unduly influenced by redundant or noisy attributes. To address this, we implement a distance metric over raw attributes, which ensures that our final meta-representation accurately captures both the properties of individual instances and the dataset.

The main challenge lies in designing an adaptive metric compatible with heterogeneous tasks. In this paper, we first formulate the distance measure in the following form:

$$\text{dist}(\boldsymbol{x}_i, \boldsymbol{x}_j) = \left( \sum_{k=1}^{d} w_k \cdot |\boldsymbol{x}_{ik} - \boldsymbol{x}_{jk}|^p \right)^{\frac{1}{p}} , \tag{6}$$

where $\boldsymbol{x}_{ik}$ denote the $k$-th dimension of $\boldsymbol{x}_i$. We set $p \in \{1, 2\}$ and $w_k > 0$ is a weight for each dimension. When $w_k = 1$, the distance in Equation 6 degenerates to the Euclidean distance ($p = 2$) or Manhattan distance ($p = 1$). Given the training set $\mathcal{D} = \{\boldsymbol{X}, \boldsymbol{Y}\}$ of a dataset where $\boldsymbol{X}$ and $\boldsymbol{Y}$ denote the instance matrix and label vector, respectively, we derive feature weights from the mutual information shared between individual attributes and their labels

$$w_k = \text{normalize}\left( \text{MI}(\boldsymbol{X}_{:k}, \boldsymbol{Y}) \right) . \tag{7}$$

$\boldsymbol{X}_{:k}$ is the $k$-th column of $\boldsymbol{X}$, *i.e.*, the vector containing values of the $k$-th attribute of all instances. $\text{MI}(\cdot, \cdot)$ calculates the mutual information between two sets, which measures the dependency between an attribute and the labels (Brown et al., 2012). The larger the mutual information, the more important an attribute is, so that we increase its weight in Equation 6. The normalize$(\cdot)$ normalizes input values by dividing their cumulative sum. The experiments validate that integrating this distance metric in meta-representation significantly enhances the model's generalization ability.

**Meta-representation in the few-shot scenario.** The previously discussed meta-representation assumes there are at least $K$ neighbors in the set $\mathcal{D}_{y=c}$. However, in some applications like few-shot classification or the existence of minority class, $\mathcal{D}_{y=c}$ might only contain a limited number of prototypes smaller than $K$. To address the data scarcity challenge, we pad the meta-representation with its last value (the largest distance) (Yang & Gopal, 2012). We also investigate augmenting the instances to size $K$ by swapping their attributes (Ucar et al., 2021a). In detail, the $k$-th element in $\boldsymbol{x}_i$ is replaced by another value in $\boldsymbol{X}_{:k}$ *with the same label* for a given probability. This supervised swap strategy mimics the distribution of the training data in a particular class. In experiments, both strategies help in few-shot scenarios, why the former one performs better, so we set the padding strategy as our default choice.

**Remarks.** The notion of meta-representation has been previously leveraged in multi-label learning to express the relationship between an instance and a specific label, which facilitates decoupling the label correlations (Yang & Gopal, 2012; Zhang & Wu, 2015). The meta-representation is also used together with the raw tabular features to assist text clasification (Canuto et al., 2014; 2018). In this paper, we provide a more general form of meta-representation. We employ meta-representation as a pivotal tool to construct a pre-trained model that can effectively operate across multiple heterogeneous tabular datasets. We also emphasize the metric-based variant and the strategies to deal with few-shot scenarios when a deep neural network is pre-trained over the meta-representations.

## 4.2 CLASSIFICATION VIA META-REPRESENTATION

Given a dataset $\mathcal{D}$, we represent an instance $\boldsymbol{x}_i$ with $\{\phi_c(\boldsymbol{x}_i)\}_{c=1}^{C}$. Based on the meta-representation, we need to obtain the prediction score for each class in a classification task. Define the score for each class as

$$[s(\boldsymbol{x}_i)_1, \ldots, s(\boldsymbol{x}_i)_C] = \mathbf{T}_\Theta \left( [\phi_1(\boldsymbol{x}_i), \ldots, \phi_C(\boldsymbol{x}_i)] \right) . \tag{8}$$

$\mathbf{T}_\Theta$ is a transformation that captures the class membership patterns from the meta-representation for each class and then outputs the corresponding class-wise classification scores. $\Theta$ denotes the learnable parameters in $\mathbf{T}$. In TabPTM, we implement $\mathbf{T}$ with Multi-Layer Perceptron (MLP), *i.e.*,

$$s(\boldsymbol{x}_i)_c = \mathbf{MLP}(\phi_c(\boldsymbol{x}_i)), \ \forall c = 1, \ldots, C \ . \tag{9}$$

When multiple types of distances are used, we concatenate them together at first and then use Equation 9 to map the concatenated meta-representation vectors to a scalar. We also apply Transformer (Vaswani et al., 2017) based on the output of MLP to correlate the predictions of $C$ classes.

$$[s(\boldsymbol{x}_i)_1, \ldots, s(\boldsymbol{x}_i)_C] = \mathbf{Transformer}\left([\mathbf{MLP}(\phi_1(\boldsymbol{x}_i)), \ldots, \mathbf{MLP}(\phi_C(\boldsymbol{x}_i))]\right) \ . \tag{10}$$

Transformer utilizes a self-attention mechanism and works as a set-to-set mapping. We denote this variant as TabPTM$^\dagger$, and experiments validate that the additional transformer is necessary when the size of the downstream dataset is large. We use an additional projection in Transformer to map the output class-specific vectors into scalars. The detailed architectures of **MLP** and **Transformer** are described in Appendix A. Based on the scores, the predicted class for $\boldsymbol{x}_i$ is

$$\hat{y}_i = \arg\max_c \ \{s(\boldsymbol{x}_i)_1, \ldots, s(\boldsymbol{x}_i)_C\} \ . \tag{11}$$

The classification strategy based on meta-representation fits heterogeneous tasks with different attributes and class spaces. Therefore, the meta-representation-based classification enables the usage of a joint model over heterogeneous tasks.

### 4.3 Pre-training with Meta-Representation

Based on the previous discussions, we pre-train a joint model, *i.e.*, the transformation $\mathbf{T}_\Theta$, whose parameters are shared across multiple seen tabular datasets.

$$\min_\Theta \ \sum_{t=1}^{T} \sum_{i=1}^{N_t} \ell\left(\mathbf{T}_\Theta(\{\phi_c(\boldsymbol{x}_i^t)\}_{c=1}^{C_t}), \ y_i^t\right) \ . \tag{12}$$

The transformation $\mathbf{T}_\Theta$, pre-trained across $T$ datasets, links the meta-representation to the final classification score. Given a downstream dataset $\mathcal{D}_u$, we first obtain the meta-representation for each instance, and the learned $\mathbf{T}_\Theta$ could be applied directly without additional fine-tuning. In other words, the pre-trained model $\mathbf{T}_\Theta$ in TabPTM is able to generalize across datasets in a *training-free* manner. The detailed pre-training and deployment workflows of TabPTM are summarized in Algorithm 1 and Algorithm 2 in the appendix.

## 5 Experiments

In this section, we first describe the experimental setups. Then we validate the generalization ability of our TabPTM in various unseen datasets with different configurations. Finally, we analyze the effectiveness of our TabPTM as well as meta-representation through ablation studies.

### 5.1 Setups

**Datasets**. We collect 22 open-source real-world tabular datasets from various fields, including medical, software, and speech recognition domains. We first split 12 relatively larger datasets into parts, one is used as seen datasets for pre-training and another part is used as downstream datasets. 10 remaining medical datasets are also selected as unseen datasets, together with the previous six datasets, to assess the downstream generalization performance of our pre-trained model. The detailed statistics of all datasets are listed in Table 4. We show the results of this partition in the main paper and the results we re-split all datasets in Appendix D. For each dataset, we randomly sample 80% of them as the training set, and the remaining 20% instances are used for test. In the training set, we randomly hold out 20% of training instances as the validation set.

**Evaluation criteria**. After pre-training on all seen datasets, we evaluate the classification accuracy of the model on unseen datasets. There are two configurations. In the full-shot scenario, we keep the whole training set for a downstream dataset, and the average accuracy over 10 random seeds is

Table 1: Average accuracy on 10 unseen datasets in the medical domain. The whole training set of each dataset is used. The best results are shown in bold. TabTPM utilizes MLP to implement the model, while TabPTM[†] incorporates Transformer to further correlate the class-wise predictions. TabPTM variants make predictions *without training*.

| | SVM | XGBoost | MLP | FT-T | TabCaps | DANets | TabPFN | XTab | DEN | TabPTM | TabPTM[†] |
|---|---|---|---|---|---|---|---|---|---|---|---|
| BC | 67.24 | 68.10 | 64.48 | 65.17 | 67.93 | 67.59 | 67.59 | 66.55 | 63.62 | **68.79** | 67.93 |
| BW | 97.14 | 97.23 | 96.64 | 97.07 | 96.36 | 97.64 | 97.14 | 97.50 | 96.71 | **99.29** | 98.57 |
| BWD | 97.37 | 96.23 | 96.32 | 97.26 | 97.02 | **97.64** | 97.15 | 96.14 | 94.74 | 95.61 | 96.49 |
| ECD | 77.78 | 78.89 | 77.41 | 75.19 | 79.63 | 82.96 | 77.78 | 83.07 | 78.89 | 84.07 | **85.19** |
| HC | 52.46 | 53.11 | 51.64 | 52.30 | 52.30 | 53.77 | **53.44** | 48.36 | 51.80 | 51.80 | 51.94 |
| HH | 81.36 | 83.05 | 82.88 | 78.64 | 81.36 | **83.39** | 81.02 | 83.22 | 78.47 | 79.66 | 80.51 |
| HV | 30.00 | 34.50 | 34.50 | 29.25 | 34.00 | 35.00 | 30.00 | 32.00 | 28.75 | **36.50** | 33.00 |
| HOC | 85.14 | **87.84** | 82.57 | 83.51 | 83.24 | 79.05 | 83.78 | 71.49 | 66.89 | 85.68 | 86.08 |
| MAM | 81.87 | 83.94 | 82.23 | 84.77 | 83.99 | 83.32 | **84.61** | 83.89 | 75.39 | 82.38 | 83.16 |
| SPE | 67.92 | 63.40 | 68.68 | 68.87 | 68.30 | 63.58 | **70.94** | 70.00 | 64.34 | 70.19 | 70.00 |
| **MEAN** | 73.82 | 74.63 | 73.74 | 73.20 | 74.41 | 74.39 | 74.35 | 73.22 | 69.96 | **75.40** | 75.29 |

reported (Gorishniy et al., 2021). While in the few-shot scenario, we randomly select $\{5, 10, 20, 40\}$ training instances per class from each dataset to evaluate whether the tabular model is able to deal with the limited size of training data. For each few-shot configuration, we sample 50 times and report the average results. Due to the page limit, we leave the standard deviation values in Appendix C.

**Comparison methods**. We compare TabPTM with three types of methods. First are the classical tabular classification methods, such as Support Vector Machine (SVM) and XGBoost (Chen & Guestrin, 2016). The second part contains deep tabular models, such as Multi-Layer Perceptron (MLP) (Kadra et al., 2021), FT-Transformer (FT-T) (Gorishniy et al., 2021), TabPFN (Hollmann et al., 2023), TabCaps (Chen et al., 2023a), DANets (Chen et al., 2022). The third part involves methods that fine-tune a pre-trained model on downstream datasets, such as XTab (Zhu et al., 2023) and Distribution Embedding Networks (DEN) (Liu et al., 2022b).

**Implementation details**. We implement our model with a five-layer MLP and a three-layer Transformer for the variant of TabPTM. During the pre-training, we set the learning rate as 0.001 and randomly sample 1024 examples from a seen dataset in each iteration. For the first two groups of comparison methods, we tune their hyper-parameters and carry out early stopping on the corresponding validation set of a given dataset. For the last type of comparison methods and ours, we use a model's average accuracy over all validation sets of the seen datasets to tune the hyper-parameters.

## 5.2 GENERALIZATION ABILITY OF THE PRE-TRAINED MODEL

**Full-shot accuracy on unseen datasets**. The average comparison accuracy results are reported in Table 1 and Table 2. The former contains relatively smaller datasets in the medical domain, while the latter contains relatively larger datasets with sizes larger than 5000. Notably, our TabPTM variants make predictions for a downstream dataset instance *without additional training, rather than tuning dataset-specific hyper-parameters separately like other classical and deep tabular models*. We use TabPTM to denote the basic version using MLP to implement the classifier, while TabPTM[†] denotes the variant applying transformer to correlate the class-wise predictions.

Based on the results, we find that the classical ensemble approach XGBoost achieves good results in most datasets, especially when the size of the training set is larger. Our TabPTM achieves the best performance in 3 of the 10 datasets in Table 1, and the two variants obtain the best two average results over the 10 medical downstream datasets. When the dataset becomes larger in Table 2, the additional transformer in TabPTM[†] helps more, and TabPTM variants achieve promising results with very short training/prediction time (300 times faster than XGBoost).

The deep tabular baseline MLP is trained over raw features for each downstream dataset separately. Although our TabPTM also adopts the MLP architecture, it is clear that the pre-trained model extracts the sharable knowledge across datasets and with better generalization ability, especially when the dataset is relatively small. XTab learns a joint transformer module during pre-training, and the

Table 2: Average accuracy on 6 relatively larger unseen datasets (size > 5000). The whole training set of each dataset is used. We omit TabPFN since it cannot deal with larger datasets. TabPTM variants make predictions *without training*. In addition to the mean value over the six datasets, the last row lists each method's average training and prediction time (in seconds) over these downstream datasets. The running time is evaluated on a system equipped with an Intel(R) Xeon(R) Silver 4210R CPU @ 2.40GHz, 376GB RAM, and one NVIDIA RTX-3090 GPU.

| | SVM | XGBoost | MLP | FT-T | TabCaps | DANets | XTab | DEN | TabPTM | TabPTM$^\dagger$ |
|---|---|---|---|---|---|---|---|---|---|---|
| churn | 85.25 | **85.99** | 85.66 | 85.92 | 85.61 | 85.34 | 85.60 | 72.48 | 85.45 | 85.45 |
| crowd | 42.00 | **47.17** | 43.53 | 39.80 | 45.83 | 46.57 | 42.73 | 35.13 | 44.47 | 44.97 |
| eye | 56.35 | **72.36** | 60.98 | 62.87 | 58.15 | 57.93 | 56.55 | 43.04 | 61.94 | 62.22 |
| htru | 97.96 | **98.11** | 98.09 | 98.09 | 98.03 | 97.94 | 98.05 | 94.18 | 97.94 | 97.96 |
| jm1 | 81.21 | **81.62** | 81.03 | 81.87 | 80.90 | 80.82 | 81.03 | 80.49 | 77.53 | 80.88 |
| satellite | 99.31 | **99.41** | 99.29 | 99.15 | 99.03 | 99.06 | 99.13 | 99.03 | 97.52 | 99.25 |
| MEAN | 77.01 | **80.78** | 78.10 | 76.54 | 77.93 | 77.94 | 77.18 | 70.73 | 77.48 | 78.46 |
| Time (s) | 2.3 $\times 10^2$ | 1.8$\times 10^3$ | 8.7$\times 10^3$ | 4.7$\times 10^3$ | 5.4 $\times 10^2$ | 8.0$\times 10^3$ | 3.2$\times 10^2$ | 7.7 $\times 10^2$ | **5.7** | 6.2 |

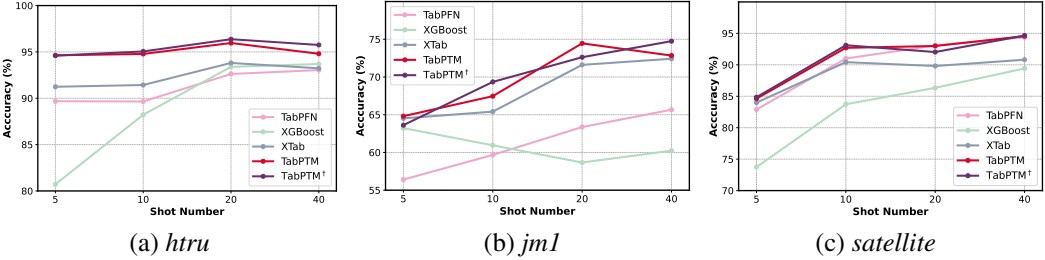

(a) *htru*          (b) *jm1*          (c) *satellite*

Figure 3: The average results of few-shot classification over multiple trials. For each downstream dataset, $\{5, 10, 20, 40\}$ examples per class (shot) are randomly sampled as the training set.

remaining parameters of a model are further fine-tuned for each downstream task. Our TabPTM outperforms XTab in 9 of the 16 datasets without additional training of its parameters.

The full-shot evaluation results indicate that our TabPTM variants make *fast yet accurate* predictions for various types of heterogeneous downstream datasets. The TabPTM could be a good choice when the model deployment efficiency is emphasized in some real-world applications.

**Few-shot accuracy on unseen datasets**. We further investigate whether TabPTM also keeps its superiority when the training set is very small. In this few-shot evaluation, we show the change in the classification accuracy when the number of instances per class (shot) increases in $\{5, 10, 20, 40\}$. We mainly compare with XGBoost, the few-shot tabular model TabPFN, and the pre-training tabular approach XTab. Due to the limited number of training instances, we use their default hyper-parameters. The few-shot results of three datasets in Table 2 are shown in Figure 3. The few-shot classification results show that TabPTM variants outperform others, verifying its few-shot generalization ability. For example, the 5-shot accuracy of TabPTM is ˜5% higher than TabPFN/XGBoost on *htru* dataset.

## 5.3 ANALYSIS OF META REPRESENTATION

**Discriminative ability of the metric-based meta-representation**. We further analyze the effectiveness of the meta-representation and the learned TabPTM. We first show the TSNE visualization of the raw attributes and the metric-based meta-representation on two datasets in Figure 4. We use different colors/shapes to denote different classes. We find the meta-representation makes the features discriminative on some datasets, which means the class-wise similarity distribution reveals the intrinsic property of a tabular dataset.

Based on the meta-representation, we further show whether the jointly learned MLP helps. We tune some basic classifiers, such as $k$NN and XGBoost, on the metric-based meta-representation for each dataset separately and compare them with our pre-trained MLP in TabPTM. The results are listed in Table 3. Comparing with results in Table 1, We find that $k$NN gets higher results than DANets and XTab on HOC, which validates the effectiveness of the metric-based meta-representation. However,

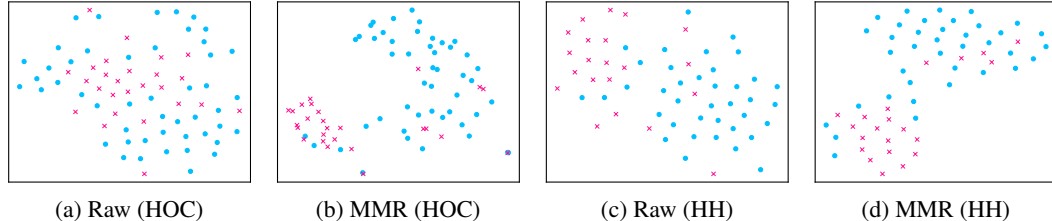

| (a) Raw (HOC) | (b) MMR (HOC) | (c) Raw (HH) | (d) MMR (HH) |

Figure 4: The TSNE visualization over horse-colic (HOC) and heart-hungarian (HH) datasets, based on their raw features (Raw) and their metric-based meta-representation (MMR). The blue 'o' and pink 'x' denote different classes. The metric-based meta-representation improves the discriminative ability of the datasets to some extent.

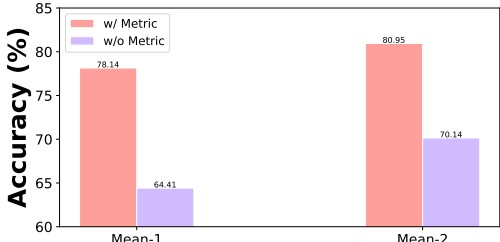

Figure 5: The average accuracy over 10 medical downstream datasets (Mean-1) and six relatively larger datasets (Mean-2) when we use meta-representation with or without the distance metric. The results clearly demonstrate that the metric is necessary in constructing meta-representation as well as a generalizable TabPTM model.

Figure 6: The average accuracy over 10 medical downstream datasets (Mean-1) and six relatively larger datasets (Mean-2) when TabPTM variants predict in a training-free manner. We also show the results when we Fine-Tune (FT) the learned TabPTM variants over each downstream dataset separately with a fixed number of iterations.

applying XGBoost on the meta-representation cannot achieve as good results as the pre-trained MLP in TabPTM. The superiority of TabPTM over the independently trained MLP also indicates that TabPTM extracts useful classification ability across multiple pre-trained datasets.

**The influence of the metric on meta-representation**. We compare the results when we pre-train TabPTM over the vanilla meta-representation and its metric-based variant in Figure 5. The results clearly indicate that the distance metric filters out redundant and noisy attributes, which is necessary to improve the generalization ability of TabPTM.

**Will additional fine-tuning help?** In Figure 6, we show the change in results when we execute additional fine-tuning over the pre-trained TabPTM. In detail, using the pre-trained model as initialization, we apply 30 iterations of gradient descent on each downstream dataset. The results show that the training-free version performs well on smaller datasets, while the fine-tuned version helps on relatively larger datasets.

Table 3: The classification accuracy on HH and HOC datasets, as well as the mean accuracy over 10 medical datasets. The classical classifiers are tuned based on the metric-based meta-representation.

|      | $k$NN | XGBoost | MLP | TabPTM |
|------|-------|---------|-----|--------|
| HH   | 78.14 | 64.41   | 79.03 | **79.66** |
| HOC  | 80.95 | 70.14   | 85.27 | **85.68** |
| MEAN | 70.87 | 64.47   | 73.91 | **75.40** |

## 6 CONCLUSION

Considering the large amount of heterogeneous tabular datasets in many machine learning fields, we explore a way to pre-train a "foundation model" and extend its generalization ability to downstream datasets. We address the primary challenge of disparate attribute and class spaces across datasets with the usage of metric-based meta-representation. Our pre-trained TabPTM can be directly applied to unseen datasets in a training-free manner. The model achieves competitive performance in various scenarios, which acts as an effective solution for practical tabular data applications.

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

There are four parts in the appendix:

- More details of our TabPTM approach in Appendix A;
- Details of the experimental setups in Appendix B;
- The whole experimental results including standard deviation in Appendix C;
- Additional ablation studies on TabPTM in Appendix D.

## APPENDIX A    DISCUSSION AND DETAILS ON OUR APPROACH

### A.1    DETAILS OF THE SCORE TRANSFORMATION

In Equation 8, we utilize a transformation $\mathbf{T}_\Theta$ to map the meta-representation to the prediction score of an instance, for all $C$ classes. The transformation could be implemented via various kinds of deep neural networks.

We describe the detailed architectures of the deep model following (Gorishniy et al., 2021). Multi-Layer Perceptron (MLP) contains several layers of non-linear blocks

$$\mathbf{MLP}(\boldsymbol{x}) = \text{Linear}(\text{MLPBlock}(\ldots(\text{MLPBlock}(\boldsymbol{x})))) \tag{13}$$

$$\text{MLPBlock}(\boldsymbol{x}) = \text{Dropout}(\text{ReLU}(\text{Linear}(\boldsymbol{x}))) . \tag{14}$$

The Linear block means a fully connected layer with linear projection. MLP maps the class-wise meta-representation $\phi_c(\boldsymbol{x}_i)$ to the prediction score $s(\boldsymbol{x}_i)_c$ (a scalar):

$$s(\boldsymbol{x}_i)_c = \mathbf{MLP}(\phi_c(\boldsymbol{x}_i)), \ \forall c = 1, \ldots, C . \tag{15}$$

Although this is a one-to-one mapping from the class-specific meta-representation to the confidence score, we validate its effectiveness in our experiments.

The mapping $\mathbf{T}$ could also be implemented with Residual Network (ResNet) (He et al., 2016):

$$\mathbf{ResNet}(\boldsymbol{x}) = \text{Prediction}\left(\text{ResNetBlock}\left(\ldots\left(\text{ResNetBlock}\left(\text{Linear}\left(\boldsymbol{x}\right)\right)\right)\right)\right) \tag{16}$$

$$\text{ResNetBlock}(\boldsymbol{x}) = \boldsymbol{x} + \text{Dropout}(\text{Linear}(\text{Dropout}(\text{ReLU}(\text{Linear}(\text{BatchNorm}(\boldsymbol{x})))))) \tag{17}$$

$$\text{Prediction}(\boldsymbol{x}) = \text{Linear}(\text{ReLU}(\text{BatchNorm}(\boldsymbol{x}))) . \tag{18}$$

Different from MLP, ResNet has a residual link from its input to the output, and Batch Normalization (Ioffe & Szegedy, 2015) is introduced in the building block of ResNet.

Furthermore, we also implement the mapping with Transformer (Vaswani et al., 2017), which is a self-attention mechanism as a set-to-set mapping. Different from MLP and ResNet, Transformer takes a set of $L$ vectors (tokens) $\{\boldsymbol{x}_1, \ldots, \boldsymbol{x}_L\}$ as input, and outputs their corresponding transformations:

$$\mathbf{Transformer}(\{\boldsymbol{x}_1, \ldots, \boldsymbol{x}_L\}) = \text{TransformerBlock}\left(\ldots\left(\text{TransformerBlock}\left(\{\boldsymbol{x}_1, \ldots, \boldsymbol{x}_L\}\right)\right)\right)$$

$$\text{TransformerBlock}(\{\boldsymbol{x}_1, \ldots, \boldsymbol{x}_L\}) = \text{FFN}(\text{MSA}((\{\boldsymbol{x}_1, \ldots, \boldsymbol{x}_L\})))$$

$$\text{FFN}(\boldsymbol{x}) = \text{Linear}(\text{ReLU}(\text{Linear}(\boldsymbol{x}))) .$$

Specifically, the **Transformer** contains a sequential TransformerBlock. In each TransformerBlock, the set of input vectors are forwarded to multi-head self-attention layer $\text{MSA}(\cdot)$ at first, and then to another feed-forward network $\text{FFN}(\cdot)$. We set the final linear layer in the last $\text{FFN}(\cdot)$ as a projection which transforms the vector input to the scalar confidence scores.

Given a set of input $L$ vectors (tokens) $\{\boldsymbol{x}_1, \ldots, \boldsymbol{x}_L\}$, $\text{MSA}(\cdot)$ transform them into query, key, and value tokens with three sets of projections $\boldsymbol{W}_q$, $\boldsymbol{W}_k$, and $\boldsymbol{W}_v$. The input $\boldsymbol{x}_i$ is transformed to $\hat{\boldsymbol{x}}_i$ via

$$\hat{\boldsymbol{x}}_i = \boldsymbol{x}_i + \sum_{j=1}^{L} \alpha_j \boldsymbol{W}_v^\top \boldsymbol{x}_j$$

$$\alpha_j \propto \exp\left(\frac{\boldsymbol{x}_i^\top \boldsymbol{W}_Q \cdot \boldsymbol{K}}{\sqrt{d}}\right)$$

$$\boldsymbol{K} = \boldsymbol{W}_K^\top [\boldsymbol{x}_1, \ldots, \boldsymbol{x}_L] .$$

---

**Algorithm 1** Pre-training on multiple heterogeneous tabular datasets.

---

**Require:** $T$ tabular training set $\mathbb{D} = \{\mathcal{D}_1, \ldots, \mathcal{D}_T\}$, the initialized model $\mathbf{T}_\Theta$

1: **for all** iteration = 1,... **do**
2:     Sample a dataset say $\mathcal{D}_t$ from $\mathbb{D}$
3:     Sample a mini-batch with $B$ instances $\{\boldsymbol{x}_i^t, y_i^t\}_{i=1}^B$
4:     **for all** $(\boldsymbol{x}_i^t, y_i^t)$ **do**
5:         Get metric-based meta-representation $\{\phi_c(\boldsymbol{x}_i^t)\}_{c=1}^{C_t}$ based on a prototypes in $\mathcal{D}_t$
6:         Obtain the prediction score $\{\mathbf{T}_\Theta(\phi_c(\boldsymbol{x}_i^t))\}_{c=1}^{C_t}$
7:         Predict via $\hat{y}_i^t = \arg\max_c \{s(\boldsymbol{x}_i^t)_1, \ldots, s(\boldsymbol{x}_i^t)_{C_t}\}$
8:         Compute loss $\ell(\hat{y}_i^t, y_i^t)$
9:     **end for**
10:    Accumulate $B$ losses as Eq. 12
11:    Update $\Theta$ with SGD
12: **end for**
13: **return** Pre-trained model $\Theta$

---

**Algorithm 2** Apply the pre-trained model to the downstream tabular dataset.

---

**Require:** Tabular training set $\mathcal{D}_u$, test instance $\boldsymbol{x}_*^u$, the learned model $\mathbf{T}_\Theta$

1: Compute the metric-based meta-representation $\{\phi_c(\boldsymbol{x}_*^u)\}_{c=1}^{C_u}$ for $\boldsymbol{x}_*^u$
2: Obtain the prediction score $\{\mathbf{T}_\Theta(\phi_c(\boldsymbol{x}_*^u))\}_{c=1}^{C_u}$
3: Predict via $\hat{y}_*^u = \arg\max_c \{s(\boldsymbol{x}_*^u)_1, \ldots, s(\boldsymbol{x}_*^u)_{C_u}\}$
4: **return** The predicted label of $\boldsymbol{x}_*^u$

---

In other words, the projected input calculates its similarity with others, and its output is the weighted average of the projected values. $d$ is the dimensionality of the input tokens.

We investigate two strategies to implement $\mathbf{T}$ with Transformer. First, we consider each input as a 2-dimension vector (one for each distance value), and the Transformer summarizes all values in meta-representation. Since the Transformer module cannot differentiate the order of the input, we add positional encoding following (Vaswani et al., 2017) to indicate the distance value in the meta-representation in ascending order. In addition, we also consider applying a transformer over the output of the class-wise vectors mapped by MLP as in Equation 10, where the Transformer correlates the results of different vectors. In experiments, the latter choice works better, so we use this choice in TabPTM[†].

## A.2   THE PRE-TRAINING AND DOWNSTREAM WORKFLOW

The pre-training objective in Equation 12 is optimized in a stochastic way. In each iteration, we randomly select a seen tabular dataset and randomly sample a mini-batch from the dataset. For each sampled instance, we first calculate the metric-based meta-representation, and then make predictions with Equation 11. We summarize the two phases, *i.e.*, pre-training the tabular model and applying the learned model on the downstream dataset, in Algorithm 1 and Algorithm 2, respectively.

We also investigate a kind of data augmentation in Appendix D, where the meta-representation is calculated based on a randomly sampled class-wise training set during training, and the whole training set in the downstream task is utilized in the evaluation. Such a data augmentation can make further improvements in some cases.

## APPENDIX B   DETAILS OF EXPERIMENTAL SETUPS

### B.1   DATASETS

We experiment with 22 tabular datasets. The statistics of all the datasets are listed in Table 4. We use $C$ and $N$ to denote the class number and the instance number of the datasets. There are two types of attributes with numerical and categorical values, and we denote their numbers as "Num."

Table 4: The detailed statistics of all tabular datasets. "Abbr." means the abbreviation of the name of the tabular dataset. The $C$ and $N$ denote the class number and the instance number of the datasets. There are two types of attributes with numerical and categorical values, and we denote their numbers as "Num." and "Cat.", respectively.

| Name | Abbr. | Task type | $C$ | $N$ | Num. | Cat. | Source |
|---|---|---|---|---|---|---|---|
| accelerometer | AC | binclass | 2 | 31991 | 6 | 0 | UCI |
| amazon | AM | binclass | 2 | 32769 | 9 | 0 | OpenML |
| credit | CE | binclass | 2 | 30000 | 20 | 3 | OpenML |
| gesture | GE | multiclass | 5 | 9873 | 32 | 0 | OpenML |
| mozilla4 | MZ | binclass | 2 | 15545 | 5 | 0 | OpenML |
| phoneme | PH | binclass | 2 | 5404 | 5 | 0 | OpenML |
| churn | CH | binclass | 2 | 10000 | 10 | 1 | OpenML |
| crowd | CO | multiclass | 6 | 10845 | 28 | 0 | UCI |
| eye | EY | multiclass | 3 | 10936 | 26 | 0 | OpenML |
| htru | HT | binclass | 2 | 17898 | 8 | 0 | UCI |
| jm1 | JM | binclass | 2 | 10885 | 21 | 0 | OpenML |
| satellite | ST | binclass | 2 | 5100 | 21 | 0 | OpenML |
| breast-cancer | BC | binclass | 2 | 286 | 9 | 0 | UCI |
| breast-cancer-wisc | BW | binclass | 2 | 699 | 9 | 0 | UCI |
| breast-cancer-wisc-diag | BWD | binclass | 2 | 569 | 30 | 0 | UCI |
| echocardiogram | ECD | binclass | 2 | 131 | 10 | 0 | UCI |
| heart-cleveland | HC | multiclass | 5 | 303 | 13 | 0 | UCI |
| heart-hungarian | HH | binclass | 2 | 294 | 12 | 0 | UCI |
| heart-va | HV | multiclass | 5 | 200 | 12 | 0 | UCI |
| horse-colic | HOC | binclass | 2 | 368 | 25 | 0 | UCI |
| mammographic | MAM | binclass | 2 | 961 | 5 | 0 | UCI |
| spect | SPE | binclass | 2 | 265 | 22 | 0 | UCI |

and "Cat.", respectively. The datasets are collected from UCI Machine Learning Repository [2] or OpenML [3].

The final 10 datasets are collected from the medical domain with relatively smaller size, so we use them as the downstream task. The first 12 datasets are relatively larger (with sizes larger than 5000), so we randomly split them into two sets, each with six datasets. The experiments in the main paper utilize the first six datasets to pre-train TabPTM, and the remaining six ones as downstream datasets. In other words, we pre-train the model using six heterogeneous tabular datasets, and evaluate its generalization ability on 16 datasets with different sizes.

We also evaluate other configurations of the pre-training and downstream split. For example, we use the six datasets in the second part to pre-train the model and the first six ones as the downstream datasets. The additional results are reported in Table 12.

## B.2 ADDITIONAL IMPLEMENTATION DETAILS

We compare our TabPTM with different types of methods and describe the detailed way we tune their hyper-parameters in this subsection. We mainly follow the setups in (Gorishniy et al., 2021) to determine the hyper-parameters.

**Classical methods and deep tabular methods**. Both classical tabular methods (SVM, XGBoost) and standard deep methods (MLP) are trained for each dataset separately. We use the official hyper-parameter search spaces for deep tabular methods (FT-T, TabCaps, DANets, and TabPFN). We tune their hyper-parameters and carry out early stopping on the corresponding validation set of a given

---

[2] https://archive.ics.uci.edu/

[3] https://www.openml.org/

Table 5: Average accuracy on 10 unseen datasets. The whole training set of each dataset is used. The best results are shown in bold. TabPTM utilizes MLP to implement the model, while TabPTM$^{\dagger}$ incorporates Transformer to further correlate the class-wise predictions. Our TabPTM makes predictions *without training*. The std of TabPTM also comes from the estimation of mutual information when constructing the metric-based meta-representation.

|      | SVM | XGBoost | MLP | FTT | TabCaps | DANets | TabPFN | XTab | DEN | TabPTM | TabPTM$^{\dagger}$ |
|------|-----|---------|-----|-----|---------|--------|--------|------|-----|--------|---------|
| BC   | 67.24 | 68.10 | 64.48 | 65.17 | 67.93 | 67.59 | 67.59 | 66.55 | 63.62 | **68.79** | 67.93 |
|      | ± 0.14 | ± 0.12 | ± 0.24 | ± 0.72 | ± 0.28 | ± 0.65 | ± 0.69 | ± 0.19 | ± 4.39 | ± 0.21 | ± 0.12 |
| BW   | 97.14 | 97.23 | 96.64 | 97.07 | 96.36 | 97.64 | 97.14 | 97.50 | 96.71 | **99.29** | 98.57 |
|      | ± 0.02 | ± 0.01 | ± 0.05 | ± 0.01 | ± 0.01 | ± 0.03 | ± 0.01 | ± 0.01 | ± 0.86 | ± 0.02 | ± 0.01 |
| BWD  | 97.37 | 96.23 | 96.32 | 97.26 | 97.02 | **97.64** | 97.15 | 96.14 | 94.74 | 95.61 | 96.49 |
|      | ± 0.01 | ± 0.94 | ± 0.25 | ± 1.25 | ± 0.05 | ± 0.04 | ± 0.01 | ± 0.25 | ± 0.68 | ± 0.05 | ± 0.15 |
| ECD  | 77.78 | 78.89 | 77.41 | 75.19 | 79.63 | 82.96 | 77.78 | 83.07 | 78.89 | 84.07 | **85.19** |
|      | ± 0.01 | ± 0.04 | ± 0.01 | ± 0.06 | ± 0.02 | ± 0.15 | ± 0.01 | ± 0.2 | ± 3.33 | ± 0.15 | ± 0.84 |
| HC   | 52.46 | 53.11 | 51.64 | 52.30 | 52.30 | 53.77 | **53.44** | 48.36 | 66.89 | 51.80 | 51.94 |
|      | ± 0.94 | ± 2.16 | ± 1.05 | ± 1.26 | ± 0.98 | ± 1.85 | ± 1.67 | ± 2.16 | ± 2.12 | ± 1.25 | ± 2.51 |
| HH   | 81.36 | 83.05 | 82.88 | 78.64 | 81.36 | **83.39** | 81.02 | 83.22 | 78.47 | 79.66 | 80.51 |
|      | ± 0.25 | ± 1.62 | ± 1.24 | ± 2.95 | ± 1.56 | ± 0.82 | ± 1.83 | ± 1.52 | ± 3.11 | ± 1.07 | ± 1.25 |
| HV   | 30.00 | 34.50 | 34.50 | 29.25 | 34.00 | 35.00 | 30.00 | 32.00 | 28.75 | **36.50** | 33.00 |
|      | ± 1.25 | ± 0.85 | ± 1.77 | ± 2.14 | ± 1.56 | ± 2.56 | ± 3.54 | ± 1.56 | ± 3.25 | ± 1.56 | ± 2.84 |
| HOC  | 85.14 | **87.84** | 82.57 | 83.51 | 83.24 | 79.05 | 83.78 | 71.49 | 66.89 | 85.68 | 86.08 |
|      | ± 0.02 | ± 0.04 | ± 0.24 | ± 0.36 | ± 0.05 | ± 0.36 | ± 0.01 | ± 0.05 | ± 2.12 | ± 0.37 | ± 0.01 |
| MAM  | 81.87 | 83.94 | 82.23 | 84.77 | 83.99 | 83.32 | **84.61** | 83.89 | 75.39 | 82.38 | 83.16 |
|      | ± 0.82 | ± 0.15 | ± 0.14 | ± 0.21 | ± 0.63 | ± 0.27 | ± 0.33 | ± 0.53 | ± 2.14 | ± 0.46 | ± 0.13 |
| SPE  | 67.92 | 63.40 | 68.68 | 68.87 | 68.3 | 63.58 | 70.94 | 70.00 | 64.34 | 70.19 | 70.00 |
|      | ± 0.52 | ± 1.05 | ± 1.21 | ± 1.82 | ± 0.59 | ± 1.2 | ± 1.25 | ± 1.84 | ± 2.73 | ± 1.05 | ± 0.92 |
| MEAN | 73.82 | 74.63 | 73.74 | 73.2 | 74.41 | 74.39 | 74.35 | 73.22 | 69.96 | 75.40 | 75.29 |
|      | ± 0.39 | ± 0.69 | ± 0.62 | ± 1.07 | ± 0.57 | ± 0.79 | ± 0.93 | ± 0.83 | ± 2.47 | ± 0.61 | ± 0.87 |

dataset. All hyper-parameters are selected by Optuna library[4] with Bayesian optimization over 30 trials. The best hyper-parameters are used and the average accuracy over 10 different random seeds is calculated.

**Pre-training and fine-tuning approaches**. For XTab, We reuse the checkpoint with the highest number of training epochs from the official implementation, then we perform evaluations on the target datasets using XTab's light fine-tuning approach. For DEN, we divide all pre-training datasets into binary and multiclass groups. Each group is then used to train models on their corresponding downstream unseen datasets. We set the learning rate as 0.001 and fine-tune the transform block on the downstream tasks. When we fine-tune TabPTM in Figure 6, we set the learning rate as 0.001 and fine-tune the whole model for 30 epochs.

**Training-free approaches**. We implement our model with a five-layer MLP or a five-layer ResNet. The Transformer-based TabPTM$^{\dagger}$ consists of a two-layer MLP and an additional three-layer Transformer. During the pre-training, we randomly sample 1024 examples from a seen dataset in each iteration.

## APPENDIX C   WHOLE EXPERIMENTAL RESULTS

The full results including average accuracy and standard deviation of Table 1, Table 2, and Figure 3 are listed in Table 5 and Table 6.

## APPENDIX D   ADDITIONAL ABLATION STUDIES

We analyze the properties of TabPTM from the following aspects.

---

[4]https://optuna.org/

Table 6: The average results of few-shot classification on *htru*, *jm1*, and *satellite*. $\{5, 10, 20, 40\}$ examples per class are randomly sampled as the training set.

| *htru* | TabPFN | XGBoost | XTab | TabPTM | TabPTM$^\dagger$ |
|---|---|---|---|---|---|
| 5-shot | 89.68$\pm$ 0.86 | 80.72$\pm$ 0.56 | 91.24$\pm$ 0.95 | **94.63** $\pm$ 0.74 | 94.60$\pm$ 0.88 |
| 10-shot | 89.65$\pm$ 0.82 | 88.22$\pm$ 0.62 | 91.43$\pm$ 0.74 | 94.78$\pm$ 0.36 | **95.05**$\pm$ 0.34 |
| 20-shot | 92.62$\pm$ 0.52 | 93.39$\pm$ 0.22 | 93.81$\pm$ 0.71 | 95.96$\pm$ 0.43 | **96.37**$\pm$ 0.25 |
| 40-shot | 93.04$\pm$ 0.25 | 93.70$\pm$ 0.27 | 93.23$\pm$ 0.39 | 94.80 $\pm$ 0.32 | **95.75**$\pm$ 0.25 |
| *jm1* | TabPFN | XGboost | XTab | TabPTM | TabPTM$^\dagger$ |
| 5-shot | 56.41$\pm$ 1.56 | 63.23$\pm$ 2.02 | 64.52$\pm$ 1.67 | **64.81**$\pm$ 1.53 | 63.62$\pm$ 2.29 |
| 10-shot | 59.67$\pm$ 0.78 | 60.95$\pm$ 1.14 | 65.41 $\pm$ 1.02 | 67.46$\pm$ 0.74 | **69.35**$\pm$ 1.33 |
| 20-shot | 63.37$\pm$ 0.84 | 58.67$\pm$ 1.25 | 71.61$\pm$ 0.63 | **74.46**$\pm$ 0.56 | 72.62$\pm$ 0.84 |
| 40-shot | 65.66$\pm$ 1.53 | 60.24$\pm$ 0.75 | 72.42$\pm$ 0.45 | 72.84$\pm$ 0.69 | **74.75**$\pm$ 0.94 |
| *satellite* | TabPFN | XGboost | XTab | TabPTM | TabPTM$^\dagger$ |
| 5-shot | 82.89 $\pm$ 0.79 | 73.74$\pm$ 1.83 | 83.99$\pm$ 2.46 | 84.62$\pm$ 1.24 | **84.85**$\pm$ 1.17 |
| 10-shot | 91.00 $\pm$ 0.73 | 83.72$\pm$ 0.82 | 90.42$\pm$ 1.91 | 92.70$\pm$ 0.43 | **93.11**$\pm$ 0.41 |
| 20-shot | **93.08** $\pm$ 0.75 | 86.34$\pm$ 0.35 | 89.81$\pm$ 0.79 | 92.99$\pm$ 0.42 | 92.03 $\pm$ 0.77 |
| 40-shot | 94.38$\pm$ 0.56 | 89.42$\pm$ 0.45 | 90.81$\pm$ 0.56 | 94.56$\pm$ 0.45 | **94.65**$\pm$ 0.57 |

Table 7: Average accuracy on six relatively larger unseen datasets. The whole training set of each dataset is used. Various strategies to generate prototypes are compared, which influence the generalization ability of meta-representation.

| | Random | $k$-Means | $k$-Means w/neg | TabPTM |
|---|---|---|---|---|
| churn | 74.47 | 72.04 | 73.77 | **85.45** |
| crowd | 43.03 | 34.43 | **45.17** | 44.47 |
| eye | 55.81 | 29.65 | 54.35 | **61.94** |
| htru | 96.31 | 95.97 | 96.80 | **97.94** |
| jm1 | 68.84 | **80.96** | 71.39 | 77.53 |
| satellite | 99.22 | 61.95 | **99.32** | 97.52 |
| MEAN | 72.95 | 62.50 | 73.47 | **77.48** |

## D.1 INFLUENCE OF PROTOTYPES IN META-REPRESENTATION

The meta-representation is calculated based on the similarity between an instance to the prototypes in a certain training set $\mathcal{D}_{y=c}$ as in Equation 4. We investigate various strategies to generate the prototypes and evaluate which strategy helps TabPTM.

- Random. We randomly select a subset of instances from $\mathcal{D}_{y=c}$, which keeps the majority of instances but decreases the size of the set.
- $k$-Means. We apply the clustering approach to split $\mathcal{D}_{y=c}$ into several parts, and the centers of the parts are used as prototypes.
- $k$-Means w/ Neg. Given a class $c$, in addition to consider the cluster centers in $\mathcal{D}_{y=c}$, we also consider applying $k$-Means in the set $\bigcup_{j\neq c} \mathcal{D}_{y=j}$. In other words, we treat instances from other classes as negative ones, and concatenate the distance to negative centers in the meta-representation.
- TabPTM. The whole instances in $\mathcal{D}_{y=c}$ are kept as the prototypes.

We pre-train the model with MLP transformation and evaluate their classification ability on various downstream datasets. The results are shown in Table 7. We find the Random strategy cannot work well since it loses some important details in some classes. The clustering approaches provide good results in some cases. For example, considering the distance to the negative centers works well in

Table 8: Average accuracy on six downstream datasets. We investigate a data augmentation strategy during pre-training by random sampling a proportion of instances in the selected datasets. When we set the sampling ratio as 100%, it means we do not apply augmentation and the meta-representation is the most accurate. In the main experiments, TabPTM does not use data augmentation during pre-training.

| Ratio | 70% | 80% | 90% | 100% |
|---|---|---|---|---|
| churn | 85.44 | 85.44 | 85.39 | **85.45** |
| crowd | 44.27 | 43.00 | 43.63 | **44.47** |
| eye | 60.09 | 60.69 | **62.08** | 61.94 |
| htru | 97.91 | **97.94** | **97.94** | **97.94** |
| jm1 | 76.76 | 77.14 | **78.08** | 77.53 |
| satellite | 99.31 | 99.01 | **99.36** | 97.52 |
| MEAN | 77.30 | 77.20 | **77.75** | 77.48 |

Table 9: Average accuracy on three downstream datasets. Different sizes ($\{3,6,9\}$) of pre-training tabular datasets are considered, which influence the generalization ability of meta-representation.

| | 3 datasets | 6 datasets | 9 datasets |
|---|---|---|---|
| churn | 85.44 | **85.45** | **85.45** |
| eye | 61.83 | 61.94 | **62.01** |
| satellite | 97.45 | 97.52 | **99.41** |
| MEAN | 81.57 | 81.64 | **82.29** |

"crowd" and "satellite". However, the performance of the clustering-based variants is not stable, so we choose to keep the whole $\mathcal{D}_{y=c}$ in TabPTM.

As shown in Algorithm 1, we sample mini-batches from a selected dataset in each iteration. Since the meta-representation is calculated based on the similarity between an instance and the prototypes, we use the "Random" strategy as a kind of data augmentation during pre-training. In detail, we randomly sample a proportion of datasets in each iteration, so that the meta-representation is noisy since it cannot access the whole data. While given a downstream dataset, we consider the whole training set to obtain the meta-representation. The results are listed in Table 8. We find such a random prototype sampling strategy improves the downstream classification ability, especially when we set the sampling ratio as 90%.

## D.2 INFLUENCE OF THE SIZE OF PRE-TRAINING DATASETS

Six heterogeneous tabular datasets are used for pre-training in our main experiments. We analyze how the size of tabular datasets influences the generalization ability of TabPTM.

Since there are 12 larger tabular datasets in total, we randomly select three of them as the downstream datasets, and select three (*mozilla4*, *credit*, *gesture*), six (*mozilla4*, *credit*, *gesture*, *amazon*, *accelerometer*, *phoneme*), and nine (*mozilla4*, *credit*, *gesture*, *amazon*, *accelerometer*, *phoneme*, *htru*, *jm1*, *crowd*) datasets in the remaining part for pre-training TabPTM.

The accuracy values over three downstream datasets are reported in Table 9. Based on the same set of downstream datasets, all pre-trained models are compared in a fair manner. We find with larger pre-training datasets, the training-free generalization ability of TabPTM improves. We conjecture by pre-training on a huge number of heterogeneous tabular datasets, the "foundation" tabular models could be obtained.

Table 10: Average accuracy on three downstream datasets. Different dimension values $K$ of meta-representation are used to pre-train TabPTM.

|  | 16 | 32 | 64 | 128 | 256 |
|---|---|---|---|---|---|
| churn | 85.31 | 85.45 | **85.5** | 85.45 | 85.38 |
| crowd | 44.9 | **44.97** | 44.9 | 44.47 | 43.67 |
| eye | **63.66** | 63.06 | 62.53 | 61.94 | 60.44 |
| htru | **98.03** | 97.98 | 97.94 | 97.94 | 97.95 |
| jm1 | 80.68 | 80.82 | 76.47 | 77.53 | **81.14** |
| Satellite | 99.22 | 99.42 | 97.03 | **97.52** | 99.22 |
| MEAN | **78.63** | 78.62 | 77.4 | 77.48 | 77.97 |

Table 11: The average accuracy over 10 medical downstream datasets (Mean-1) and 6 relatively larger datasets (Mean-2) with different implementations of the transformation **T** in TabPTM.

|  | Mean-1 | Mean-2 |
|---|---|---|
| MLP | **75.40** | 77.48 |
| ResNet | 74.25 | 77.14 |
| Transformer | 75.29 | **78.10** |

## D.3 THE INFLUENCE OF THE DIMENSION OF THE META-REPRESENTATION

In Equation 5, we consider the nearest $K$ prototypes in the training set of each class, which makes the meta-representation with dimension $K$. We set $K = 128$ by default in previous experiments. We pre-train TabPTM over meta-representations with different dimensions $K$ in Table 10. We find different downstream datasets may prefer various dimension values.

## D.4 META-REPRESENTATION WITH OTHER TOP-LAYER ARCHITECTURES

We also investigate other implementations of the mapping **T** from meta-representation to the classification scores. Since using the vanilla Transformer cannot generalize well, we show the results when we use MLP, ResNet, and an additional Transformer over the output of MLP in Table 11. The results indicate that MLP is a good choice in most cases, and the additional Transformer helps when the size of datasets are larger.

## D.5 RESULTS WITH OTHER PRE-TRAINING DATASETS

Recall that there are 22 tabular datasets. In addition to the 10 tabular datasets from the medical domain used as the downstream datasets, we randomly select six of the remaining 12 datasets as the pre-training ones and six of them as the downstream ones. We exchange the pre-training and downstream datasets in this subsection. The results are shown in Table 12. Our TabPTM still achieves the best average accuracy over the 10 downstream datasets. The results validate the training-free generalization ability of TabPTM variants.

Table 12: Average accuracy on 10 unseen datasets. We use another set of six datasets as the pre-training datasets w.r.t. Table 1. The whole training set of each dataset is used. The best results are shown in bold. TabTPM utilizes MLP to implement the model, while TabPTM[†] incorporates Transformer to further correlate the class-wise predictions. Our TabPTM makes predictions *without training*.

|      | SVM   | XGBoost | MLP   | FT-T  | TabCaps | DANets | TabPFN | XTab  | DEN   | TabPTM  | TabPTM[†] |
|------|-------|---------|-------|-------|---------|--------|--------|-------|-------|---------|-----------|
| BC   | 67.24 | 68.10   | 64.48 | 65.17 | 67.93   | 67.59  | 67.59  | 66.55 | 65.34 | **68.97** | 67.24   |
| BW   | 97.14 | 97.23   | 96.64 | 97.07 | 96.36   | 97.64  | 97.14  | 97.5  | 96.00 | **97.93** | 97.14   |
| BWD  | 97.37 | 96.23   | 96.32 | 97.26 | 97.02   | 97.64  | 97.15  | 96.14 | 92.81 | 97.46     | **98.25** |
| ECD  | 77.78 | 78.89   | 77.41 | 75.19 | 79.63   | 82.96  | 77.78  | 83.07 | 77.04 | 82.22     | **85.19** |
| HC   | 52.46 | 53.11   | 51.64 | 52.30 | 52.30   | 53.77  | 53.44  | 48.36 | 51.80 | **55.90** | 51.64   |
| HH   | 81.36 | 83.05   | 82.88 | 78.64 | 81.36   | **83.39** | 81.02 | 83.22 | 78.47 | 75.59   | 81.53     |
| HV   | 30.00 | 34.50   | 34.50 | 29.25 | 34.00   | **35.00** | 30.00 | 32.00 | 27.50 | 33.25   | 29.25     |
| HOC  | 85.14 | **87.84** | 82.57 | 83.51 | 83.24  | 79.05  | 83.78  | 71.49 | 72.39 | 85.94     | 86.22     |
| MAM  | 81.87 | 83.94   | 82.23 | 84.77 | 83.99   | 83.32  | **84.61** | 83.89 | 53.42 | 82.90   | 83.11     |
| SPE  | 67.92 | 63.40   | 68.68 | 68.87 | 68.30   | 63.58  | **70.94** | 70.00 | 65.85 | 70.00   | 69.43     |
| MEAN | 73.82 | 74.63   | 73.74 | 73.20 | 74.41   | 74.39  | 74.35  | 73.22 | 68.06 | **75.02** | 74.90   |

