# OpenReview forum: "Training-Free Generalization on Heterogeneous Tabular Data via Meta-Representation"
_ICLR.cc/2024/Conference — Submitted to ICLR 2024_

### Official Review · Reviewer_jAb7 · 2023-10-16

**Soundness:** 2 fair
**Presentation:** 3 good
**Contribution:** 3 good
**Rating:** 5
**Confidence:** 4

**Summary:**

This paper introduces a novel approach to enable training-free generalization for tabular datasets.

The core idea is something like:

For any given dataset, the input label data (x, y) is restructured into a new format: (distance to prototypes of class c, likelihood of the label of class c). This uniform data representation allows different datasets to be organized in a consistent manner. Thus, a model trained on this standardized format can effectively generalize across various tabular datasets.

On unseen datasets, the proposed model achieves superior performances and saves training time.

**Strengths:**

- It is exciting to witness a tabular learning model deliver remarkable performance without the necessity for fine-tuning.

- On the whole, the model is sound and the meta-representation extraction is novel.

**Weaknesses:**

The dataset used in this study is somewhat limited. Although I have confidence in the model's ability to generalize effectively to new datasets by representing data points in terms of their similarity to prototypes, there are concerns about its adaptability to other datasets. The reason is straightforward: the issue of feature heterogeneity remains partly unresolved, as there may not be a latent state that aligns to all decision boundaries with similarity measure. The top-K operation on prototypes helps to mitigate this problem to some extent but doesn't completely address it. For example, any transformations applied to the features within the tables, like replacing a feature 'x' with '1/x' (such transformation is reversible, and the information has not been altered or lost), might make the hidden states fail. Therefore, I suspect that this model might only exhibit robustness on certain datasets, and it would be beneficial for the authors to clarify this point and clarify about when the model is effective and when it is not. Otherwise, this paper may potentially lead to misinterpretations in subsequent research.

It might be also advisable to consider using benchmark datasets like those in Taptap (https://arxiv.org/pdf/2305.09696.pdf) or Grinsztajn's work (https://proceedings.neurips.cc/paper_files/paper/2022/file/0378c7692da36807bdec87ab043cdadc-Supplemental-Datasets_and_Benchmarks.pdf) to enhance the credibility of your finding. Perhaps incorporating the average rank and rank standard deviation (like XTab) can provide deeper insights into this research.

**Questions:**

See equations (3) and (4), is each instance in $D_{y=c}$ considered a different prototype?

There exist multiple versions of DANet, each differing in the number of layers (e.g., DANet-8, DANet-27). You should specify the layer configuration in this paper.

---

> ### Author Response · Authors · 2023-11-22
> **Respond to Reviewer jAb7 (part 1)**
>
> The reviewer's constructive comments are highly valued and appreciated. Due to space constraints, our response will be divided into two parts.
>
> **Q1**: The issue of feature heterogeneity remains partly unresolved, as there may not be a latent state that aligns to all decision boundaries with similarity measure. The top-K operation on prototypes helps to mitigate this problem to some extent but doesn't completely address it. For example, any transformations applied to the features within the tables, like replacing a feature 'x' with '1/x' (such transformation is reversible, and the information has not been altered or lost), might make the hidden states fail.
>
> **A1**: Thanks. We will respond from three aspects.
>
> First, the feature heterogeneity is the main difficulty in training a model over multiple tabular datasets. Some existing methods deal with the problem by padding the feature, learning dimension-agnostic transformations, and using large language models. The uniform representation in this paper only encodes the common states of different models, while may lose the specific information.
>
> Second, for the given example, if we replace a feature 'x' with '1/x' (such transformation is reversible), the semantic meaning of this feature is changed. While in TabPTM, the mutual information between the feature and the label remains stable, and the feature will be emphasized in calculating the meta representation.
>
> Third, it is the relationship between instances, i.e., the distance between features, that determines the values in meta-representation. If two samples are close to each other under positive feature x, it will still be close when we reverse the feature. The meta-representation is robust to part of the transformations in practice.
>
> **Q2**: This model might only exhibit robustness on certain datasets, and it would be beneficial for the authors to clarify this point and clarify when the model is effective and when it is not. Use benchmark datasets.
>
> **A2**: Thanks for the suggestions. In addition to the classification results in the paper, we validate the robustness of the model in regression tasks, based on the benchmark datasets in the mentioned paper.
>
> In a regression task, the label $y$ of an instance $\boldsymbol{x}_i$ is a continuous value, so that we modify the meta representation of an instance $\phi(\boldsymbol{x}_i)$ as the concatenation of its distance with neighbours as well as their corresponding labels. In detail, if the $K$ nearest neighbours (based on the Euclidean distance) of instance $\boldsymbol{x}_i$ is ${\boldsymbol{x}_1, \dots, \boldsymbol{x}_K}$, we define the meta representation for regression as:
>
> $\phi(\boldsymbol{x}_i) = [dist(\boldsymbol{x}_i, \boldsymbol{x}_1), \ldots, dist(\boldsymbol{x}_i, \boldsymbol{x}_j), \ldots, dist(\boldsymbol{x}_i, \boldsymbol{x}_K), y_1, \ldots, y_j, \ldots, y_K] \in \mathbb{R}^{2K}$.
>
> Then, we can use a top-layer regressor to predict the label, i.e., $\hat{y_i}=\mathbf{T}_\Theta(\phi(\boldsymbol{x}_i))$.
>
> If $\mathbf{T}_\Theta$ is a linear model, the prediction could be implemented based on a neighbourhood manner by weighted averaging the labels of the neighbours.
>
> We implement $\mathbf{T}_\Theta$ as a MLP. Similar to the protocol of classification we used in our paper, we collected 12 regression datasets. 9 of 12 datasets are sourced from Grinsztajn's benchmarks. We pre-train our TabPTM on half of them (house_sales, california_housing, MiamiHousing2016, ailerons, abalone, pole), and evaluate the learned model on the remaining datasets (analcatdata_supreme, wine_quality, house_16H, sulfur, yprop_4_1, houses).
>
> We set $K = 8$. By utilizing both Euclidean and Manhattan distance, the dimension of meta representation is $4K$. XGBoost and MLP are tuned on a single dataset. TabPTM is tuned on 6 pre-training datasets and generalizes to 6 unseen datasets without fine-tuning. The experiments are listed in the table below:
>
> |                     | XGBoost | MLP    | TabPTM |
> | ------------------- | ------- | ------ | ------ |
> | analcatdata_supreme | 0.0849  | 0.0761 | 0.0838 |
> | wine_quality        | 0.6691  | 0.4972 | 0.5787 |
> | house_16H           | 0.6576  | 0.5564 | 0.4541 |
> | sulfur              | 0.0234  | 0.0235 | 0.0199 |
> | yprop_4_1           | 0.0301  | 0.0213 | 0.0207 |
> | houses              | 0.2359  | 0.2169 | 0.2312 |
>
> We will add more datasets in the future, and analyze the robustness of TabPTM.

---

> > ### Author Response · Authors · 2023-11-22
> > **Respond to Reviewer jAb7 (part 2)**
> >
> > **Q3**: Incorporating the average rank.
> >
> > **A3**:
> >
> > The average rank results of methods in Table 1 are listed as below:
> >
> > | SVM  | XGBoost | MLP  | FT-T | TabCaps | DANets | TabPFN | XTab | DEN  | TabPTM | TabPTM$^\dagger$ |
> > | ---- | ------- | ---- | ---- | ------- | ------ | ------ | ---- | ---- | ------ | ---------------- |
> > | 6.2  | 4.6     | 7.7  | 6.8  | 5.6     | 4.2    | 4.8    | 6.2  | 9.8  | 4.5    | 4.5              |
> >
> > Thanks for the suggestion. We will incorporate the average rank into the evaluation of the methods.
> >
> > **Q4**: See equations (3) and (4), is each instance in $D_{y=c}$ considered a different prototype?
> >
> > **A4**: Keeping the full training set as in equation (3) is beneficial in most cases. When there is a memory limitation, we randomly sample 10K instances per class from the training set as prototypes.
> >
> > **Q5**: Specify the number of layers in DANet.
> >
> > **A5**: Thanks. We use the default number of layers in the official code, i.e., DANet-20.
> >
> > Thanks again for the suggestions!

---

> ### Comment · Reviewer_jAb7 · 2023-11-23
>
> Hi, thank you for your answer.
>
> Your approach appears to solely rely on the distance to neighboring data, but it discards the direction. Could you explain (experiments are not necessary) on the potential issues arising from discarding direction? I am a bit curious about this because intuitively, I feel that direction might be crucial, although it could introduce significant computational costs.
>
> Additionally, on irregular decision boundaries, situations arise where samples with shorter distances may have completely different labels. This suggests potential theoretical shortcomings in the method. I guess if we encode the entire dataset, relying solely on distance is acceptable. However, in cases where the dataset is extensive, and encoding the entire dataset is impractical, or when there are changes in data distribution (or some data becomes inaccessible) in reference, challenges may arise. Is this correct?

---

> > ### Author Response · Authors · 2023-11-23
> >
> > Thanks for your feedback. As the rebuttal period is concluding, we would like to provide a brief response.
> >
> > - In the representation theorem, it is possible to construct a classifier using a kernel-like approach.
> >
> > - The similarity of samples relies on the choice of distance metric, and a good metric should bring similar samples closer, mitigating the mentioned issues. Currently, we have only experimented with Lp-distance to demonstrate the effectiveness of the framework.

---

### Official Review · Reviewer_65fa · 2023-10-27

**Soundness:** 1 poor
**Presentation:** 2 fair
**Contribution:** 1 poor
**Rating:** 1
**Confidence:** 4

**Summary:**

This paper addresses the difficult problem of multi-datasets pre-training/fine-tuning/few-shots-learning where the main challenge is the lack of a common and coherent attributes representation between the different datasets.
In order to solve this problem, in a classification setting, the authors propose :

1- to extract a representative instance x_c of each class of each dataset

2- to represent each instance x of a dataset by the increasing list of distances to the K nearest class-representative instances of this dataset

3- to use this kind of "spectral K-nn" representation as a universal feature replacement

Based on this class-aware representation, they train a generic model to predict the class for any dataset.
The experiments then show that this approach is able to predict correctly the class "without further training" on a fresh dataset.

This paper could be entitled "using k-nearest class-representative neighbours to encode multi-table data"

**Strengths:**

- The paper is clear
- According to the experiments the method seems to work relatively well

**Weaknesses:**

I recommand rejection because :

- The "related work section" does not cover the huge literature on k-nearest neighbors and their multiple metric-based variants. It would require at least half a page to position this work amongst these methods
- The authors claim that their algorithm is "training-free". This may appear as a philosophical question, but to my opinion selecting class-representative items in a dataset is a -- strong and costly -- form of training, just as selecting the support vectors for a SVM is a form a training.
- To my opinion this proposal is merely shifting the "not-that-hard" classification problem to the "not-that-easy" metric-learning and representative-choice problems.

**Questions:**

If we are to tain a k-nn or a metric-based classification algorithm for each new dataset, wouldn't be the class-probability outputs of more robust classifiers like XGBoost or SVM be better and cheaper "universal representations" ?

---

> ### Author Response · Authors · 2023-11-22
> **Respond to Reviewer 65fa**
>
> Thanks for the valuable feedback the reviewer have shared with us.
>
> **Q1**: The "related work section" does not cover k-nearest neighbors and metric-based variants.
>
> **A1**: Since the target of this paper is to learn a joint model over multiple heterogeneous tabular datasets, we do not emphasize metric-based methods a lot since the kNN or other metric-based methods cannot be applied across datasets. We discuss the relationship between ours and other meta-feature methods in "remarks" in Section 4.1. We will add more discussions in the related work part in the final version.
>
> **Q2**: Selecting class-representative items in a dataset is a -- strong and costly -- form of training, just as selecting the support vectors for a SVM is a form a training. To my opinion, this proposal is merely shifting the "not-that-hard" classification problem to the "not-that-easy" metric-learning and representative-choice problems.
>
> **A2**: Thanks. In the paper, we first describe the idea of learning a general model with meta-representation, and the meta-representation aims to encode the relationship between an instance and its nearest prototypes in a particular class. We have tried various strategies, and in our experiments, we set *the whole training set* of a class as the prototype set. In other words, all training instances in a class are treated as prototypes. Therefore, the computational burden mainly lies in the similarity search, which could be easily accelerated in practical implementations. We show the training time comparison between various models, including the non-deep method and deep methods, in Table 2. The results indicate the training-free method makes predictions on a target dataset fast.
>
> **Q3**: The class-probability outputs of more robust classifiers like XGBoost or SVM are better and cheaper "universal representations" ?
>
> **A3**: We can use the prediction results as the representation of an instance based on some robust classifier. However, there are several difficulties.
>
> First, it depends on the training quality of the classifier, and sometimes, it requires careful tuning of the hyper-parameters. As demonstrated in Table 1, XGBoost and SVM can not exhibit particularly outstanding performance even with tuned hyper-parameters.
>
> In addition, using XGBoost and SVM also requires training time (as shown in Table 2). So such a kind of representation cannot be obtained in a "cheaper" way.
>
> Furthermore, since different datasets have different sizes of classes, the "universal representations" *have different dimensions*, which impedes the sharing of knowledge across datasets.
>
> The main idea of this paper is to learn a general model over multiple heterogeneous datasets, and TabPTM represents instances in the same form.
>
> Thanks again for the comments!

---

> > ### Comment · Reviewer_65fa · 2023-11-23
> > **Thanks for your answers**
> >
> > I will however maintain that encoding instances by their ordered distances to the nearest class prototypes (over the whole dataset!) and stacking a Transformer neural model over this representation to predict the class is merely a compute-intensive and clumsy way of building a simple nearest-neighbour metric-based classifier.
> >
> > - As expected for a metric-based classifier, on Table 2, the TabPTM performance is shown to be very close to the SVM one, but unfortunately the cross-validation standard deviation is not provided.
> > - The training time is not the most important : inference time of metric-based classifiers if known to be prohibitive (especially if you keep the whole dataset as prototypes). The best way to save inference-time is to select these class representatives during the training phase as we do for SVM or Rocchio.
> > - The two naive types of metric-based classifiers Rocchio and k-NN with their inference time should be included as baseline in the experiment section. See https://nlp.stanford.edu/IR-book/html/htmledition/vector-space-classification-1.html

---

> ### Author Response · Authors · 2023-11-23
> **Respond to Reviewer 65fa**
>
> Thanks for your feedback. As the rebuttal period is concluding, we would like to provide a brief response.
>
> - We have explored sampling the dataset instead of using the whole dataset. During training, a sampling approach was employed, where only a subset of the dataset was sampled in each iteration (as shown in Table 8). During inference, the entire dataset was utilized. Efficient algorithms for nearest neighbor search exist. The problem is a common challenge for all metric-based methods.
>
> - We have experimented with these two simple baselines, but they did not yield satisfactory results. Thank you for pointing out, and we will consider incorporating them in the future.

---

### Official Review · Reviewer_bXAG · 2023-10-30

**Soundness:** 3 good
**Presentation:** 3 good
**Contribution:** 1 poor
**Rating:** 3
**Confidence:** 4

**Summary:**

The paper introduces tabular data pre-training through meta-representation, enabling pre-training on a collection of heterogeneous tabular datasets. This model can be directly applied to previously unseen datasets with diverse attributes and classes without the need for additional training. Specifically, the model computes the distance between a record for which we want to have a representation and the other records within the same class. Then the model selects the K smallest distance values, where the resulting vector of length K forms the meta-representation $\phi_c(x_i)$. For classification tasks, one can train a mapping between the representation vectors and the classification scores using a combination of MLP and Transformer. The authors demonstrate the effectiveness of the proposed model using 22 real-world datasets.

**Strengths:**

The proposed approach does not requires additional training for unseen datasets.

**Weaknesses:**

1. The main idea of the approach is a distance-based measurement between data records. However, it may seem somewhat trivial because the Lp norm-based metric (it appears that the authors used L1 or L2 norm) forms an Lp space, and this may be insufficient to represent tabular data, which exhibits complex distributions.


2. Two heterogeneous tables can have different label distributions, even if feature columns have similar distributions between the two tables. Due to this inconsistent characteristic, fitting one table's distribution to that of another table may not work well.


3. I believe you should clarify the claim, whether the proposed model is for extracting representations or classifying data labels. If the model includes both calculating the representation $\phi$ and training $\textbf{T}_{\theta}$, then the model should be a classifier, not just a representation method. This is because only with the proposed representation $\phi_c$, one cannot directly perform other tasks.


4. The experimental results are insufficient to demonstrate the superiority of the proposed method.

**Questions:**

As I understand, all the classes in all datasets are learnt as they are each individual class. Then how do you classify unseen data where for which classes are also unknown?

---

> ### Author Response · Authors · 2023-11-22
> **Respond to Reviewer bXAG**
>
> We are grateful for the valuable suggestions.
>
> **Q1**: Lp space may be insufficient to represent tabular data.
>
> **A1**: Thanks. Since diverse tabular datasets have different sets of features and classes, we investigate how to represent all the tabular datasets in a *uniform* form to support the training of a shared model. In our meta-representation, distances between one instance and its same-class nearest neighbours are utilized. We show the general idea of meta-representation and its basic form by using the L1 and L2 distance. Our experiments indicate that learning with a shared model can make up for information loss using the Lp norm-based metric. Furthermore, we can also concatenate other kinds of distance in meta-representation to improve its representation ability.
>
> **Q2**: Two heterogeneous tables can have different label distributions, fitting one table's distribution to that of another table may not work well.
>
> **A2**: That's right. The main challenge of learning a shared model over multiple datasets lies in the fact they have heterogeneous feature and label distributions.
>
> The main idea of using meta-representation to learn a general model across datasets is to capture the *intrinsic classification characteristic* over datasets. The meta-representation encodes the similarity distribution of a class. If an instance belongs to a class with high probability (e.g., near the center of a class or in the high-density area of the class), most distance values to its nearest neighbours are small. In contrast, the distance value between an instance and the non-target classes should have larger values. The top-layer module $T_\Theta$ captures such a kind of similarity distribution for classification, which could be shared across datasets. We also illustrate this idea in Figure 1 in the paper, and a discussion of what could be shared across datasets in Section 4.2.
>
> **Q3**: Whether the proposed model is for extracting representations or classifying data labels.
>
> **A3**: Thanks. Our model TabPTM is a classification model based on the meta-representation. The meta-representation formulates an instance into a $K$-dimensional form no matter how many dimensions it originally had. Meanwhile, the model $T_{\Theta}$ captures the shared pattern (the pattern in the similarity distribution in the last answer) across datasets.
>
> **Q4**: The experimental results are insufficient to demonstrate the superiority of the proposed method.
>
> **A4**: Firstly, the baseline methods we compare are tuned specifically on individual datasets, making them more prone to achieving good performance, with computational costs increasing linearly as the dataset size grows. However, TabPTM is a model applied directly to all downstream tasks without the need for additional fine-tuning. Secondly, our TabPTM achieves the best performance in 3 of the 10 datasets in Table 1, and the two variants obtain the best two average results over the 10 downstream datasets. In Table 2, TabPTM achieves promising results with very short training/prediction time (300 times faster than XGBoost). We also validate the effectiveness of TabPTM on regression tasks (as shown on Respond to Reviewer XUXC A1).
>
> **Q5**: How to classify unseen data for which classes are also unknown?
>
> **A5**: In our experiments, we evaluate our TabPTM on datasets with *unseen* classes and features, and the results validate the ability of TabPTM over datasets with unseen classes.
>
> Given a new dataset, we calculate the meta-representation of an instance w.r.t. each one of the unseen classes. For example, given $C$ unseen classes, we obtain $C$ meta-representations with $K$-dimension. Then we apply the learned model $T_\Theta$ over them to obtain the $C$-dimensional class confidence vector without additional training. The instance is classified as one of the unseen classes by selecting the index with the maximum confidence.
>
> Thanks again for the feedback!

---

### Official Review · Reviewer_XUXC · 2023-11-01

**Soundness:** 2 fair
**Presentation:** 3 good
**Contribution:** 3 good
**Rating:** 6
**Confidence:** 4

**Summary:**

The paper proposes to classify tabular data by learning a shared model above a meta-representation. Samples are represented as distances to prototypes (K the closest samples of particular classes in terms of weighted p-norm distance). The proposed approach TabPTM displays training free generalization to unseen dataset in both few shot and full size regimes and performs comparably to a set of classical and deep-learning baselines.

**Strengths:**

- The paper is well written
- The goal of learning a shared model for a set of heterogeneous tabular problems is hard and intriguing
- The proposed approach is simple and seems to outperform simple baselines and perform on par with more advanced DL methods, in a training free transfer setting

**Weaknesses:**

- Regression tasks which are plentiful in practice in tabular problems are not supported by the method.
- Description of the tuning and evaluation protocols are not sufficient for reproducibility. Please provide a more detailed info on how you tuned and evaluated the baselines.
- Some important baselines are missing. Proposed method uses augmentations applicable to other models, but MLP, Transformers and DANets are evaluated without augmentations (which might help in both low-shot and small data regimes)
- TabPFNs do support datasets with around 10k samples (and TabPFN performance was shown to improve with more samples in the original paper). I believe TabPFN should be added to the comparison on larger datasets.
- Number of datasets for which the model is evaluated is rather small by today's standards in tabular DL, especially given that the method generalizes without training. It would be great if you could also add results on the benchmark from `[1]`

**References**
- `[1]` Grinsztajn, Léo, Edouard Oyallon, and Gaël Varoquaux. "Why do tree-based models still outperform deep learning on typical tabular data?." Advances in Neural Information Processing Systems 35 (2022): 507-520.

**Questions:**

- What are the problem with supporting regression with a similar approach?
- Could you describe the tuning and evaluation protocols for the baselines in more detail?

Minor remarks:
- ",why the former one" -- while? (Meta-representation in the few-shot scenario subsection)

---

> ### Author Response · Authors · 2023-11-22
> **Respond to Reviewer XUXC (part 1)**
>
> Thanks very much for the valuable suggestions provided by the reviewer. Due to space constraints, our response will be divided into two parts.
>
> **Q1**: Supporting regression with a similar approach.
>
> **A1**: Thanks for the suggestion. The meta representation of an instance encodes its distance with same-class nearest neighbours, and the idea could be extended to the regression scenario.
> In a regression task, the label $y$ of an instance $\boldsymbol{x}_i$ is a continuous value, so that we modify the meta representation of an instance $\phi(\boldsymbol{x}_i)$ as the concatenation of its distance with neighbours as well as their corresponding labels. In detail, if the $K$ nearest neighbours (based on the Euclidean distance) of instance $\boldsymbol{x}_i$ is ${\boldsymbol{x}_1, \dots, \boldsymbol{x}_K}$, we define the meta representation for regression as:
>
> $\phi(\boldsymbol{x}_i) = [dist(\boldsymbol{x}_i, \boldsymbol{x}_1), \ldots, dist(\boldsymbol{x}_i, \boldsymbol{x}_j), \ldots, dist(\boldsymbol{x}_i, \boldsymbol{x}_K), y_1, \ldots, y_j, \ldots, y_K] \in \mathbb{R}^{2K}$.
>
> Then, we can use a top-layer regressor to predict the label, i.e., $\hat{y_i}=\mathbf{T}_\Theta(\phi(\boldsymbol{x}_i))$.
>
> If $\mathbf{T}_\Theta$ is a linear model, the prediction could be implemented based on a neighbourhood manner by weighted averaging the labels of the neighbours.
>
> We implement $\mathbf{T}_\Theta$ as an MLP, and experiments validate its effectiveness to become a general regression model across datasets. Similar to the protocol of classification we used in our paper, we collect 12 regression datasets, pre-training our TabPTM on half of them (house_sales, california_housing, MiamiHousing2016, ailerons, abalone, pole), and evaluating the learned model on the remaining datasets (analcatdata_supreme, wine_quality, house_16H, sulfur, yprop_4_1, houses). It is notable that these two sets of datasets do not share the same set of features or regression ranges.
>
> We set $K = 8$. By tilizing both Euclidean and Manhattan distance, the dimension of meta representation is $4K$. XGBoost and MLP are tuned on each dataset separately. TabPTM is tuned on 6 pre-training datasets and generalizes to 6 unseen datasets without fine-tuning. The experiments are listed in the table below:
>
> |                     | XGBoost | MLP    | TabPTM |
> | ------------------- | ------- | ------ | ------ |
> | analcatdata_supreme | 0.0849  | 0.0761 | 0.0838 |
> | wine_quality        | 0.6691  | 0.4972 | 0.5787 |
> | house_16H           | 0.6576  | 0.5564 | 0.4541 |
> | sulfur              | 0.0234  | 0.0235 | 0.0199 |
> | yprop_4_1           | 0.0301  | 0.0213 | 0.0207 |
> | houses              | 0.2359  | 0.2169 | 0.2312 |
>
> TabPTM obtains the best performance on half of the datasets, and the second-best performance on the remaining datasets in a training-free manner.
>
> **Q2**: Provide more detailed info on how to tune and evaluate the baselines.
>
> **A2**: We described the implementation details, including the hyper-parameter tuning, in Section B.2. For some recently proposed methods such as TabCaps, we conducted hyperparameter tuning using their official code. We will provide more details in the final version. For example, hyper-parameters of XGBoost and MLP are selected by Optuna library with Bayesian optimization over 30 trials. Hyper-parameters of TabCaps are selected by hyperopt library over 30 trials. The hyperparameter spaces are as follows:
>
> - XGBoost: {'model': {'alpha': ['?loguniform', 0, 1e-08, 100.0], 'colsample_bylevel': ['uniform', 0.5, 1.0],'colsample_bytree': ['uniform', 0.5, 1.0], 'gamma': ['?loguniform', 0, 1e-08, 100.0], 'lambda': ['?loguniform', 0, 1e-08, 100.0], 'learning_rate': ['loguniform', 1e-05, 1], 'max_depth': ['int', 3, 10],'min_child_weight': ['loguniform', 1e-08, 100000.0], 'subsample': ['uniform', 0.5, 1.0]}}
> - MLP: {"model": {"d_layers": ["$mlp_d_layers", 1, 5, 1, 512], "dropout": ["?uniform", 0.0, 0.0, 0.5 ], },"training": {"lr": ["loguniform", 1e-05, 2e-3],"weight_decay": ["?loguniform", 0.0, 1e-06, 1e-3]}}
> - TabCaps: {'lr': uniform('lr', 4e-2, 2e-1), 'sub_class': randint('sub_class', 1, 5), 'init_dim': randint('init_dim', 32, 128), 'primary_capsule_size': randint('primary_capsule_size', 4, 32), 'digit_capsule_size': randint('digit_capsule_size', 4, 32), 'leaves': randint('leaves', 16, 64)}
>
> **Q3**: Proposed method uses augmentations applicable to other models, but MLP, Transformers and DANets are evaluated without augmentations.
>
> **A3**: We investigated several strategies to obtain a $K$ dimensional meta-representation when the number of examples per class is limited. Data augmentation is one of the strategies we have tried, and finally, we find padding the meta representation with its last value performs the best, which is mentioned in Section 4.1 ("..., so we set the padding strategy as our default choice"). In our main experiments, the comparisons with other methods are fair.

---

> ### Author Response · Authors · 2023-11-22
> **Respond to Reviewer XUXC (part 2)**
>
> **Q4**: TabPFN should be added to the comparison on larger datasets.
>
> **A4**: It is mentioned in the TabPFN paper that the method can only support small-scale tabular datasets. In our experiments, most large datasets contain over 10K samples. We do try TabPFN on larger datasets, while there exist computational issues on some datasets. Thanks for the suggestion. We also noticed that there are some improved variants of TabPFN recently, and we will compare some of them in the final version.
>
> **Q5**: Add results on the benchmark from [1].
>
> **A5**: There are 22 classification datasets and 35 regression datasets in [1]. In our paper, we evaluate the results of TabPTM on 22 classification datasets with varying scale (some additional results are provided in Table 12 in Appendix d). Thanks for the suggestions. We evaluate TabPTM on 12 regression datasets (9 of the 12 datasets are sourced from benchmarks [1]), as described in Q1&A1. We will include additional datasets in the future.
>
> We appreciate the feedback provided by the reviewer.

---

### Comment · Area_Chair_KP8y · 2023-11-20
**Authors, please respond to the reviews**

Dear authors: Reviewers see positive aspects in this submission, but they do have concerns. Please submit your responses soon. Thank you!

---

> ### Author Response · Authors · 2023-11-22
> **Thanks to the AC and reviewers**
>
> Thanks to the AC for encouraging our responses. We appreciate the valuable feedback from each reviewer. We have responded to all reviewers. Hope our response may address their concerns and engage in further discussions.

---

### Meta-Review · Area_Chair_KP8y · 2023-12-14

**Metareview:**

One reviewer evaluates the paper as above the high ICLR threshold, but others do not. Personally, I tend to agree with the content of the comments of reviewer 65fa, even if his/her score is perhaps too harsh: the method suggested is a complicated variation on k-nearest neighbors and needs to be justified better from that perspective.

**Justification For Why Not Higher Score:**

Reviewers have valid criticisms.

**Justification For Why Not Lower Score:**

A future version of this paper may be able to make the case for its importance.

---

### Decision · Program_Chairs · 2024-01-16

Reject